# Positive Correlation of Serum Resistin Level with Peripheral Artery Disease in Patients with Chronic Kidney Disease Stage 3 to 5

**DOI:** 10.3390/ijerph182312746

**Published:** 2021-12-03

**Authors:** Xin-Ning Ng, Chi-Chong Tang, Chih-Hsien Wang, Jen-Pi Tsai, Bang-Gee Hsu

**Affiliations:** 1Department of Internal Medicine, Dalin Tzu Chi Hospital, Buddhist Tzu Chi Medical Foundation, Chiayi 62247, Taiwan; nancy.ngxinning@gmail.com; 2Division of Nephrology, Hualien Tzu Chi Hospital, Buddhist Tzu Chi Medical Foundation, Hualien 97004, Taiwan; dearedward1025@gmail.com (C.-C.T.); wangch33@gmail.com (C.-H.W.); 3School of Medicine, Tzu Chi University, Hualien 97004, Taiwan; 4Division of Nephrology, Dalin Tzu Chi Hospital, Buddhist Tzu Chi Medical Foundation, Chiayi 62247, Taiwan

**Keywords:** peripheral arterial disease, ankle-brachial index, chronic kidney disease, resistin

## Abstract

Chronic kidney disease (CKD) is associated with higher risk of cardiovascular disease-related ischemic events, which includes peripheral arterial disease (PAD). PAD is a strong predictor of future cardiovascular events, which can cause significant morbidity and mortality. Resistin has been found to be involved in pathological processes leading to CVD. Therefore, we aim to investigate whether resistin level is correlated with PAD in patients with non-dialysis CKD stage 3 to 5. There were 240 CKD patients enrolled in this study. Ankle-brachial index (ABI) values were measured using the automated oscillometric method. An ABI value < 0.9 defined the low ABI group. Serum levels of human resistin were determined using a commercially available enzyme immunoassay. Thirty CKD patients (12.5%) were included in the low ABI group. Patients in the low ABI group were older and had higher resistin levels as well as higher diabetes mellitus, hypertension and habit of smoking, and lower estimated glomerular filtration rate than patients in the normal ABI group. After the adjustment for factors that were significantly associated with PAD by multivariate logistic regression analysis, age and serum resistin level were independent predictors of PAD. A high serum resistin level is an independent predictor of PAD in non-dialysis CKD stage 3 to 5.

## 1. Introduction

Chronic kidney disease (CKD) is associated with higher risk of generalized atherosclerosis and cardiovascular disease (CVD)-related ischemic events [1]. Importantly, CKD, independent of conventional risk factors, is a risk factor for development of peripheral arterial disease (PAD) [2]. Data from National Health and Nutrition Examination Survey (NHANES 1999–2000) published in 2004 reported that persons with creatinine clearance <60 mL/min/1.73 m^2^ (24%) had sixfold higher prevalence of PAD than persons with a creatinine clearance >60 mL/min/1.73 m^2^ (4%) [3]. Higher prevalence rates were found in dialysis patients, as supported by data from the Dialysis Outcomes and Practice Patterns Study (DOPPS), which reported PAD prevalence rate was 25.3% [4]. Male sex, older age, diabetes, hypertension, and tobacco use were found to be significantly associated with an increased risk of PAD in dialysis patients [4,5,6].

PAD is recognized as peripheral arteries obstruction due to atherosclerosis, which is highly prevalent in patients with CKD [3,6]. Symptomatic PAD can result in decreased quality of life and could lead to significant morbidity and mortality, especially in persons with CKD [7].

Resistin is a cysteine-rich peptide consisting of 108 amino acid sequences. Resistin is involved in different metabolic processes, acting as a proinflammatory factor, resulting in production of inflammatory cytokines and increased expressions of cell adhesion molecules [8]. Emerging evidence suggests that recombinant resistin directly causes endothelial dysfunction, implying a potential role in atherosclerosis [9]. Recombinant resistin was found to induce endothelial dysfunction in vitro, due to increased expression of endothelin-1, vascular cell adhesion molecule-1, and enhanced vascular smooth muscle cells migration, ultimately contributing to atheromatous plaque formation [10]. Reilly el al. published the first important study demonstrating the relationship between atherosclerosis and resistin, by identification of increased resistin level and coronary calcium score [9]. Later on, Wang et al. found more elevated serum resistin levels in patients with acute coronary syndrome than those who were classified as control group and stable angina. Serum resistin was correlated with the number of coronary arteries with more than 50% stenosis [11]. Supporting the role of resistin in atherosclerosis, Hsu et al. further showed that elevated serum resistin was an independent predictor of PAD in patients with hypertension [12]. In addition, recent evidence suggests that resistin levels can even predict cardiovascular hospitalizations in the type 2 diabetic population with mild to moderate CKD [13]. Therefore, this study aims to investigate the correlation of resistin levels and PAD in non-dialysis CKD stage 3 to 5 patients.

## 2. Materials and Methods

### 2.1. Participants

A total of 240 non-dialysis CKD stage 3 to 5 were recruited from a medical center in Hualien, Taiwan. The recruitment occurred between January 2020 and August 2020. Blood pressure (BP) measurements were obtained in the morning, after at least 10 min of rest and measurements were obtained by a trained staff, using an appropriately sized standard mercury sphygmomanometer. The average of 3 serial measurements taken 5 min apart was used. Patients were classified as hypertensive if their blood pressure recording ≥140/90 mmHg or had been treated with one or more antihypertensive agents in the past 2 weeks. Patient with diabetes mellitus (DM) were defined as fasting plasma glucose ≥126 mg/dL or had received one or more oral antihyperglycemic drugs or insulin. This study was approved by the Institutional Review Board of Tzu Chi Hospital (IRB108-219-A). All the patients provided written informed consent before participation. Exclusion criteria were dialysis, acute infection, malignancy, stroke, amputation, heart failure, treatment with cilostazol or pentoxifylline during blood sampling, elevated ankle-brachial index (ABI) > 1.4, or refusal to provide informed consent.

### 2.2. Anthropometric Measurements

Weight was measured in minimal clothing and without shoes, recorded to the nearest 0.5 kg. Height was also recorded to the nearest 0.5 cm. Waist circumference was measured midway between the lowest ribs and the hip bones with the hands placed on the hips. Body mass index (BMI) was calculated by dividing weight (in kg) by height (in meters squared) [12].

### 2.3. Biochemical Investigations

Blood samples were obtained after overnight fasting. In each patient, approximately 5 mL blood was immediately centrifuged at 3000× *g* for 10 min and stored at 4 °C within 1 h of processing for biochemical analyses. Serum blood urea nitrogen, creatinine, fasting glucose, total cholesterol, low-density lipoprotein cholesterol, triglycerides, total calcium, and phosphorus were measured by autoanalyzer (Siemens Advia 1800; Siemens Healthcare, Henkestr, Germany). Serum resistin values were analyzed by enzyme-linked immunosorbent assay (SPI-BIO; Montigny le Bretonneux, France) [12]. The estimated glomerular filtration rate (eGFR) was calculated with the use of Chronic Kidney Disease Epidemiology Collaboration equation.

### 2.4. Assessment of ABI

An ABI-form device (VaSera VS-1000; Fukuda Denshi Co., Ltd., Tokyo, Japan) was used to measure the ABI values. Simultaneous measurement of the oscillometric blood pressures in the arms and ankles were obtained. Occlusion and monitoring cuffs were placed over all four extremities with the patient in supine position. ABI was calculated by dividing the lowest mean from the ankles by the mean in the arm. A low ABI value was defined by less than 0.9 (low ABI group). The diagnosis of PAD is established based on an ABI < 0.9 [12].

### 2.5. Statistical Analysis

The normal distribution of the results was assessed by using Kolmogorov–Smirnov test. Data with normal distribution were shown as mean ± standard deviation and comparisons between participants were conducted by using the Student’s independent t-test (two-tailed). Mann–Whitney U test was used to analyze the differences between non-normally distributed data and expressed as medians and interquartile ranges. PAD-associated variables were tested for independence by multivariate logistic regression analysis. Area under curve (AUC) was calculated using receiver operating curve (ROC) and determine serum resistin levels to predict PAD in CKD patients. Categorical variables were expressed as numbers with percentages and compared using the Chi-square test. All analyses were performed with the use of SPSS for Windows, version 19.0 (SPSS Inc., Chicago, IL, USA). A *p* value less than 0.05 was considered to indicate statistical significance.

## 3. Results

Baseline characteristics of all 240 patients with non-dialysis CKD stage 3 to 5 are summarized in Table 1. The mean age was 69.07 ± 13.67 years, the mean eGFR was 31.28 ± 15.41 mL/min, 196 patients (81.7%) had hypertension, and 86 patients (35.8%) had DM. The use of medications included angiotensin receptor blockers (ARB; *n* = 133; 55.4%), β-blockers (*n* = 70; 29.2%), calcium channel blockers (CCB; *n* = 116; 48.3%), statins (*n* = 108; 45%), and fibrate (*n* = 22; 9.2%).

A total of 30 patients with non-dialysis CKD stage 3 to 5 (12.5%) were included in the low-ABI group. There were no significant differences in gender distribution, BMI, serum lipids, creatinine, stage of CKD, coexisting glomerulonephritis, and the use of ARB, β-blockers, CCB, statin or fibrate were seen between either of the groups. Compared with participants in the control group, patients with low ABI were significantly older age (*p* < 0.001) and had more DM (*p* = 0.033), hypertension (*p* = 0.023), and had ever smoked (*p* = 0.013). Patients in the low ABI group had significantly higher fasting glucose (*p* = 0.022) and serum resistin level (*p* < 0.001), but a lower eGFR level (*p* = 0.002), as compared to normal ABI group.

Multivariate logistic regression analysis of the PAD-associated factors (DM, fasting glucose, hypertension, systolic blood pressure, ever smoker, age, eGFR, and resistin) revealed that increased serum resistin level (odds ratio [OR]: 1.141; 95% confidence interval [CI]: 1.054–1.234; *p* = 0.001), and age (OR: 1.102; 95% CI: 1.051–1.156; *p* < 0.001) were independent predictors of PAD in patients with CKD (Table 2). Furthermore, the receiver operating characteristic curve for predicting PAD revealed that the area under the curve for serum resistin was 0.699 (95% CI: 0.636–0.756, *p* = 0.0001) (Figure 1).

## 4. Discussion

This study demonstrates that non-dialysis CKD stage 3 to 5 patients with low ABI tended to be older aged, lower eGFR, and had DM, hypertension, used tobacco more and had markedly elevated serum resistin levels. Furthermore, we found that age and resistin levels were significantly associated with PAD in non-dialysis CKD stage 3 to 5 patients after we adjusted other cofounder factors.

Age, male gender, tobacco use, DM, dyslipidemia, and hypertension are well recognized as established risk factors for PAD [14,15]. The mean age in this study was 71 years, and our findings also showed correlation of DM, hypertension and tobacco use in low ABI group, consistent with previously published studies [16,17]. A Chinese cohort study demonstrated that a history of DM and tobacco use were independent risk for low ABI [16]. In the Strong Heart Study cohort, 4393 American Indians were followed for an average of 8 years, where DM, albuminuria, and hypertension were found to occur with greater frequency in the low ABI group as compared to the normal ABI group, respectively [17].

The ABI is a simple and non-invasive diagnostic tool for detection of PAD [18,19]. Although not a direct measure of PAD, a systemic review has shown that an ABI cut-off of <0.9 had a sensitivity of 97% and specificity of 89% in detecting PAD on angiography [20]. Furthermore, an ABI of <0.9 has also shown to be associated with increased cardiovascular and all-cause mortality [7,17,21]. In the Edinburgh Artery Study cohort, a low ABI was associated with an increased risk for CVD, independent of metabolic syndrome and conventional cardiovascular risk factors [22]. Li et al. prospectively evaluated Chinese patients with ABI < 0.9, and reported a 3-year all-cause mortality of 37.7% in patients with PAD and CKD, compared with only 5% in those with neither disease [16]. However, the accuracy of ABI in general population has yet to be verified in CKD populations [23].

The prevalence of PAD is greater in patients with CKD, compared with individuals with normal renal function [4]. According to the National Health and Nutrition Examination Survey 1999–2000 (NHANES), 24% of individuals with creatinine clearance <60 mL/min/1.73 m^2^ have PAD, whilst only 3.7% of individuals with creatinine clearance >60 mL/min/1.73 m^2^ have PAD [3]. Patients with both CKD and PAD are associated with significantly higher mortality rates (45%), compared to those with either condition alone (CKD alone 28%, PAD alone: 26%) [24]. Furthermore, CKD has been identified as an independent risk factor for PAD, with risk increasing with worsening renal function [2]. Despite an increased risk of cardiovascular events and mortality, PAD is still underdiagnosed [23]. Currently existing clinical care guidelines are limited by the paucity of randomized clinical trials addressing the early diagnosis and management of PAD, in CKD and end stage renal disease populations [23,25,26].

Resistin, initially described as an adipokine secreted by the adipocytes, has been suggested as a potential link between obesity, insulin resistance, and DM [27,28]. Although its expression was originally defined in adipocytes, human resistin is primarily secreted by peripheral blood mononuclear cells, macrophages, and bone marrow cells [29]. It has been shown that resistin is involved in a variety of biological processes, including the activation of innate immune systems such as nuclear factor-κB (NF-κB) and Mitogen-activated protein kinase (MAPK) pathway, by exerting proinflammatory effects [30,31]. Emerging evidence proposed that resistin plays a pathogenic role in CVD, including inflammation, endothelial dysfunction, thrombosis, and smooth muscle activity [8,9,27]. Takeishi et al. found that increased serum resistin correlated with the severity of heart failure and significantly predicts future adverse cardiac events [32]. In a prospective case-cohort study conducted by Weikert et al., 26,490 apparently healthy middle-aged subjects were followed for an average of 6 years and found individuals with highest quartile of resistin levels are related to increased risk of myocardial infarction [33]. Resistin has been shown to upregulate cytokines and adhesion molecule expression on human endothelial cells, implicating a potential role in atherosclerosis [8,10]. Gherman et al. examined the association between serum adipokine levels and PAD in a prospective case-control study, and reported that resistin levels were significantly higher in PAD patients than in the control group [34]. Furthermore, resistin levels were associated with severity of PAD, reduced amputation-free survival, and an increased rate of major adverse cardiac events, suggesting that resistin may be a novel biomarker and prognostic indicator in PAD patients [35,36]. Serum resistin levels are markedly elevated in patients with renal insufficiency, indicating a strong impact of eGFR on resistin levels, as previously reported [37]. In this regard, our findings revealed that elevated serum resistin still significantly correlated with low ABI after multivariate logistic regression analysis, even adjusted with eGFR. Our findings further showed that increased levels of resistin were found to be an independent predictor of PAD non-dialysis CKD stage 3 to 5 patients.

There are several limitations to this study that require consideration. Our study is cross-sectional in design and hence causality could not be drawn based on the current results. Second, the present study was conducted in a single center, with a specific community selected because of the high prevalence of CKD. Finally, we did not measure inflammatory markers such as C-reactive protein of these patients. Therefore, further longitudinal studies are needed to define the relationship between serum resistin and PAD in non-dialysis CKD patients before confirming the cause–effect relationship.

## 5. Conclusions

The present study showed that higher resistin level and older age were the independent predictors of PAD in non-dialysis CKD stage 3 to 5 patients. Given the potential role of resistin acting as a biomarker and therapeutic target for atherosclerosis, further studies are essential to elucidate the detailed signaling pathways and complex biological effects of resistin [38].

## Figures and Tables

**Figure 1 ijerph-18-12746-f001:**
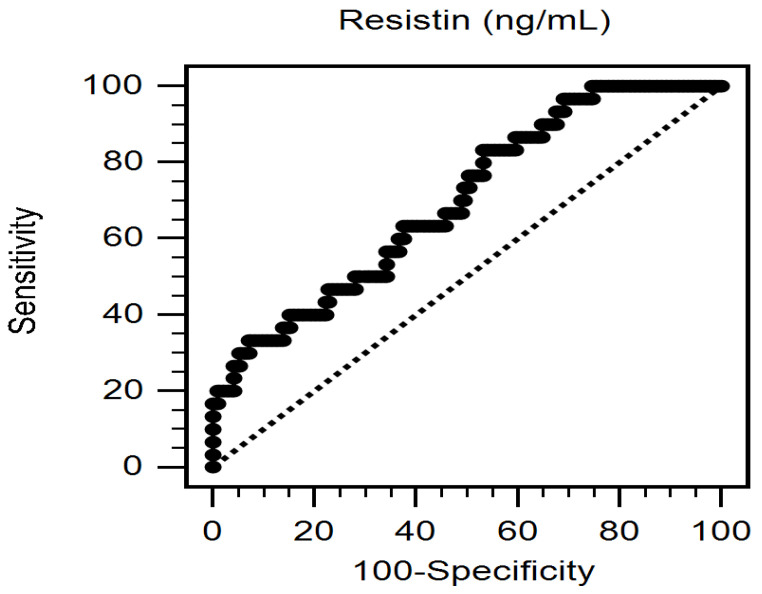
The area under the receiver operating characteristic curve indicates the diagnostic power of serum resistin levels for predicting peripheral arterial disease among 240 chronic kidney disease patients.

**Table 1 ijerph-18-12746-t001:** Clinical variables of the 240 chronic kidney disease patients with normal or low ankle brachial index group.

Characteristics	All Patients (*n* = 240)	Control Group (*n* = 210)	Low ABI Group (*n* = 30)	*p* Value
Age (years)	69.07 ± 13.67	67.64 ± 13.47	79.07± 10.68	<0.001 *
Height (cm)	158.61 ± 8.54	158.74 ± 8.55	157.73 ± 8.54	0.548
Body weight (kg)	65.84 ± 14.29	66.23 ± 13.9688	63.12 ± 16.42	0.265
Body mass index (kg/m^2^)	26.06 ± 4.64	26.16 ± 4.43	25.35 ± 5.95	0.367
Left-ankle-brachial index	1.05 (1.00–1.13)	1.07 (1.02–1.13)	0.81 (0.63–0.88)	<0.001 *
Right-ankle-brachial index	1.08 (1.02–1.13)	1.09 (1.04–1.13)	0.83 (0.66–0.89)	<0.001 *
SBP (mmHg)	147.63 ± 21.86	146.32 ± 21.19	156.83 ± 24.53	0.013 *
DBP (mmHg)	83.60 ± 12.63	83.89 ± 12.41	81.63 ± 14.11	0.362
Total cholesterol (mg/dL)	160.93 ± 41.68	159.50 ± 42.10	170.87 ± 37.73	0.163
Triglyceride (mg/dL)	121.00 (86.00–169.75)	120.00 (82.75–164.50)	130.00 (97.75–185.00)	0.248
LDL-C (mg/dL)	86.00 (69.00–112.00)	84.00 (69.00–109.25)	97.75 (69.75–139.00)	0.130
Fasting glucose (mg/dL)	100.00 (89.00–124.75)	98.00 (89.00–123.25)	110.00 (99.75–140.25)	0.022 *
Blood urea nitrogen (mg/dL)	33.00 (24.00–48.75)	33.00 (23.75–50.00)	34.50 (25.50–44.50)	0.856
Creatinine (mg/dL)	1.90 (1.40–2.78)	1.90 (1.4–2.83)	1.85 (1.40–2.35)	0.744
eGFR (mL/min)	31.28 ± 15.41	44.33 ± 25.42	33.91 ± 20.35	0.001 *
Total calcium (mg/dL)	9.14 ± 1.45	9.22 ± 1.74	8.63 ± 1.77	0.086
Phosphorus (mg/dL)	3.80 (3.30–4.20)	3.80 (3.30–4.20)	4.00 (3.10–4.33)	0.769
Resistin (ng/mL)	7.99 (6.30–12.33)	7.69 (5.91–11.80)	10.67 (7.55–22.15)	<0.001 *
Female, *n* (%)	109 (45.4)	96 (45.7)	13 (43.3)	0.806
Diabetes mellitus, *n* (%)	86 (35.8)	70 (33.3)	16 (53.3)	0.033 *
Hypertension, *n* (%)	196 (81.7)	167 (79.5)	29 (96.7)	0.023 *
Glomerulonephritis, *n* (%)	60 (25.0)	56 (26.7)	4 (13.3)	0.115
Current smoking, *n* (%)	25 (10.4)	18 (8.6)	7 (23.3)	0.013 *
ARB use, *n* (%)	133 (55.4)	114 (54.3)	19 (63.3)	0.351
β-blocker use, *n* (%)	70 (29.2)	62 (29.5)	8 (26.7)	0.745
CCB use, *n* (%)	116 (48.3)	98 (46.7)	18 (60.0)	0.172
Statin use, *n* (%)	108 (45.0)	94 (44.8)	14 (46.7)	0.844
Fibrate use, *n* (%)	22 (9.2)	20 (9.5)	2 (6.7)	0.612
CKD stage 3, *n* (%)	118 (49.2)	104 (49.5)	14 (46.7)	0.457
CKD stage 4, *n* (%)	75 (31.3)	63 (30.0)	12 (40.0)	
CKD stage 5, *n* (%)	47 (19.6)	43 (20.5)	4 (13.3)	

Values for continuous variables are given as mean ± standard deviation and tested by Student’s *t*-test; variables not normally distributed are given as median and interquartile range and tested by Mann–Whitney U test; values are presented as number (%) and analysis was performed using the chi-square test. ABI, ankle-brachial index; SBP, systolic blood pressure; DBP, diastolic blood pressure; LDL-C, low-density lipoprotein cholesterol; eGFR, estimated glomerular filtration rate; ARB, angiotensin-receptor blocker; CCB, calcium-channel blocker; CKD, chronic kidney disease. * *p* < 0.05 was considered statistically significant.

**Table 2 ijerph-18-12746-t002:** Multivariate logistic regression analysis of the factors correlated to peripheral arterial disease among 240 chronic kidney disease patients.

Variables	Odds Ratio	95% Confidence Interval	*p* Value
Resistin, 1 ng/mL	1.141	1.054–1.234	0.001 *
Age, 1 year	1.102	1.051–1.156	<0.001 *
Hypertension, present	8.335	0.885–78.496	0.064
Current smoking, present	3.017	0.776–11.726	0.111
Systolic blood pressure, 1 mmHg	1.017	0.994–1.031	0.150
Diabetes mellitus, present	2.062	0.731–5.817	0.172
Fasting glucose, 1 mg/dL	1.004	0.995–1.014	0.353
eGFR, 1 mL/min	0.997	0.964–1.031	0.853

Analysis data was performed using the multivariate logistic regression analysis (adopted factors: diabetes mellitus, hypertension, current smoking, age, systolic blood pressure, fasting glucose, eGFR, and resistin). eGFR, estimated glomerular filtration rate. * *p* < 0.05 was considered statistically significant.

## Data Availability

The data presented in this study are available on request from the corresponding author.

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
