# Peer review of "Positive Correlation of Serum Resistin Level with Peripheral Artery Disease in Patients with Chronic Kidney Disease Stage 3 to 5"

_ijerph, 2021, doi:10.3390/ijerph182312746_

Round 1

Reviewer 1 Report

The authors present results from a cross-sectional study of resistin and PAD in CKD stage 3 to 5 patients. Manuscript is well-written and results appear scientifically sound. A minor comment is for the authors to modify "PAD development" to "PAD" in the conclusion.

Reviewer 2 Report

I read with interest this article which studies 240 CKD patients, 30 of whom had a low ABI (under 0.9). They demonstrate an association between serum resistin and low ABI, however low ABI was also associated with other known factors such as age, diabetes, AH and smoking. After adjustment for confounders, age and serum resistin remained significant. The authors do an especially good job describing the role of resistin and its potential involvement in pathology. The methodology is accurately described, the results are clearly presented and the discussion addresses all the necessary points. The conclusion is supported by the data, but could be slightly extended in terms of one sentence or two of suggestios for future research. With a few exceptions described bellow, the English seems good.

Line 161: "This study demonstrates that non-dialysis CKD stage 3 to 5 patients with low ABI were tended to be older age, lower eGFR, and had DM, hypertension, tobacco used and markedly elevated serum resistin levels. Furthermore, we found that age and resistin levels were significantly associated with PAD in non-dialysis CKD stage 3 to 5 patients after adjusted other cofounder factors.

This sentence, unlike most of the rest of the manuscript, does not have to be totally correct in terms of English. I would remove the word "were", change "tobacco used" to "used tobacco more and had markedly..." or simply to "tobacco use" and change "after adjusted" to either "after we adjusted" or "after adjustment".  

Another small mistake in the Conclusions line 234: "Whether resistin act as a potential biomarker or therapeutic target for PAD, remains to be 
determined."

I believe this should be changed to "acts". I would also welcome a suggestion in the conclusion of how the authors think this could be best determined, for example what basic study design or method would be best suited for this purpose. 
